# Reducing Barriers to COVID-19 Vaccination Uptake: Community Ideas from Urban and Rural Kenya

**DOI:** 10.3390/ijerph20237093

**Published:** 2023-11-22

**Authors:** Ahmed Asa’ad Al-Aghbari, Violet Naanyu, Stanley Luchters, Eunice Irungu, Kawthar Baalawy, Till Bärnighausen, Joy Mauti

**Affiliations:** 1Heidelberg Institute of Global Health, Heidelberg University Hospital, Im Neuenheimer Feld 130.3, 69120 Heidelberg, Germany; till.baernighausen@uni-heidelberg.de (T.B.); joy.mauti@uni-heidelberg.de (J.M.); 2School of Arts and Social Sciences, Moi University, Eldoret, Kenya; vnaanyu@ampath.or.ke; 3Centre for Sexual Health and HIV AIDS Research (CeSHHAR), Harare 0002, Zimbabwe; stanley.luchters@ceshhar.co.zw; 4Department of International Public Health, Liverpool School of Tropical Medicine (LSTM), Liverpool L3 5QA, UK; 5Department of Public Health and Preventive Medicine, Ghent University, 9000 Ghent, Belgium; 6The Aga Khan Hospital Mombasa, Mumbasa, Kenya; eunice.irungu@yahoo.com (E.I.); kwthrblwy@outlook.com (K.B.)

**Keywords:** COVID-19, vaccine uptake, reduction in barriers to vaccination, health misinformation, vaccine hesitancy, Kenya

## Abstract

Following the rapid development of COVID-19 vaccines, addressing vaccine hesitancy and optimizing uptake have emerged as critical challenges, emphasizing the importance of reducing barriers toward COVID-19 vaccination. This study investigates ideas on ways to reduce barriers to COVID-19 vaccination uptake. It explores methods that can overcome COVID-19 vaccination barriers through qualitative research: interviews and group discussions involving healthcare providers, administration personnel, teachers, and individuals with chronic conditions across urban (Mombasa) and rural (Kilifi) Kenya. Audio-recorded discussions were transcribed and thematically analyzed across locations. Five themes emerged in our results regarding the reduction in barriers to COVID-19 vaccination in the context of Kenya, including awareness campaigns, engaging diverse stakeholders, using various communication techniques, capacity building to increase vaccination centers and trained staff, and lastly, revising relevant government health policies and guidelines. These results indicate the importance of adopting multiple approaches, as no single strategy can boost vaccine acceptance. Moreover, this study provides recommendations for conceiving actionable interventions to potentially boost vaccine demand and maintain routine immunization in Kenya.

## 1. Introduction

Up until the first half of 2022, the COVID-19 pandemic has infected more than 9 million people and caused more than 200,000 confirmed deaths in the continent of Africa alone, of which more than 337,000 confirmed cases and more than 5600 confirmed deaths have occurred in Kenya; we assume that those numbers are an underestimate, taking into consideration the low testing capability in Kenya [1]. One of the best methods to minimize adverse health outcomes from infection is the COVID-19 vaccine. Nonetheless, challenges to the equitable distribution and deployment of vaccines introduce a considerable barrier to accelerating the end of the pandemic.

As of August 2023, 70.5% of the world population has received at least one dose of a COVID-19 vaccine. In total, 13.49 billion doses have been administered globally, and 97,868 are now administered daily. However, only 32.5% of people in low-income countries have received at least one dose; in Kenya, precisely (as of August 2023), only 26.83% of the population have received at least one dose of the COVID-19 vaccine, of which only 17.60% were fully vaccinated according to the initial protocol in Kenya which indicates two doses to be full vaccination [2]. The enormous inequality in healthcare resources, particularly during a pandemic, undoubtedly worsens the already increasing mistrust in healthcare systems, experts, and the government, which could promote vaccine hesitancy among low- and middle-income country LMIC population [3]. Vaccine inequity remains an onerous problem globally, predominantly in low- and middle-income countries LMICs. Given their limited supply, vaccine hesitancy becomes a crucial issue to mitigate if we want to provide optimal uptake in low-resource and fragile settings [4].

The WHO has defined vaccine hesitancy as a “delay in acceptance or rejection of safe vaccines despite the availability of vaccine services” [5]. COVID-19 vaccine hesitancy is a multifaceted phenomenon influenced by a range of factors. It comprises socio-demographic, psychological, cognitive, political, and cultural aspects and varies among populations [6]. Multiple studies have demonstrated that public concern about the safety and side effects of COVID-19 vaccines, COVID-19 vaccines’ widespread misinformation, and mistrust in governments are the factors contributing to growing hesitancy toward COVID-19 vaccines [7]. However, overpowering vaccine hesitancy is tricky, and therefore, no single intervention can be resolved entirely, particularly regarding COVID-19, where scientific evidence for compelling strategies to tackle it is presently constrained [8]. Therefore, a framework to better understand the five psychological prototypes of vaccination called the 5C model was designed incorporating the following: confidence, complacency, convenience, risk calculations, and collective responsibility. This framework, developed primarily in high-income countries, harnesses the possibility of comprehending more reasonable approaches and developing strategies to manage vaccine hesitancy, but might miss elements that could be relevant to LMICs [4]. Significant information voids about vaccine hesitancy remain, particularly in Africa, due to constrained resources, specifically those that have been validated in local or regional settings [9,10].

Overcoming barriers to vaccination is a pivotal challenge both globally and particularly in Africa. Vaccine hesitancy, misinformation(infodemic), and limited access to healthcare infrastructure are significant impediments affecting immunization efforts. Economic factors and inadequate public awareness further compound the issue. Addressing these barriers, which are rooted in socio-cultural, economic, and logistical complexities, is essential for achieving higher vaccination rates and promoting public health in Africa and across the world [4,6,7]

Kenya is a low–middle-income country located in sub-Saharan Africa (SSA). Its population exceeds 50 million, with 75% living in rural areas and nearly 46% living below the poverty line. Governmental, private, and faith-based centers are the main segments of the Kenyan healthcare system. About half of healthcare is delivered in the public system in government-run establishments. By contrast, the other half is conveyed by private facilities, of which nearly 20% are faith-based organizations [11]. The direct supplier of vaccines in Kenya is the Ministry of Health. Initial guidelines, approaches, and policies associated with vaccine distribution were established according to the WHO’s Strategic Advisory Group of Experts (SAGE) Roadmap, entitling healthcare workers who have a higher risk of infection and front-line workers to access the vaccine first [12].

In various countries such as Kenya, Ethiopia, and Pakistan, research has pointed to misinformation, limited understanding, and insufficient health education as key obstacles to vaccination. These challenges can be effectively addressed by employing appropriate strategies that can manage the spread of misleading information (infodemic management), effective risk communication, and engaging with communities to enhance awareness and an understanding of vaccines. Several studies highlight the potential of these approaches to mitigate vaccination barriers and improve immunization rates [13,14,15,16,17].

To steer needed communications and engagement approaches to support the rollout of COVID-19 vaccines, it is crucial to comprehend public attitudes in the community, which is often multi-factorial and highly context-specific. This manuscript emanates from a broader study that aims to investigate public knowledge and perceptions of adult vaccination programmers, focusing on COVID-19 vaccine acceptance among adults and adolescents in rural and urban Kenya. In this particular manuscript, we focus on the ideas shared on reducing barriers to COVID-19 vaccination uptake.

## 2. Materials and Methods

### 2.1. Study Scope and Design

This qualitative study investigated public knowledge and perceptions of vaccination programs, specifically COVID-19 vaccine acceptance among adults and adolescents in Mombasa and Kilifi, Kenya. Both in-depth interviews (IDIs) and focus group discussions (FGDs) were conducted in Mombasa Old Town (Urban setting) and Rabai, Kilifi (rural setting) from 29 March to 1 April 2022.

### 2.2. Study Population, Sample, and Recruitment

This study conducted 18 IDIs and 6 FGDs (*n* = 46) in Mombasa and Kilifi Counties, with 9 IDIs and 3 FGDs at each site. The participants included health care providers (HCPs), administration personnel, teachers, and people living with chronic conditions (PLWCD) for IDIs. FGDs consisted of community members in three distinct age groups: 15–19 years, 20–49 years, and over 50 years. A purposeful selection process was used, resulting in 24 data collection sessions. In Mombasa, interviews included 2 HCPs, 2 administration personnel, 2 teachers, 2 PLWCDs, and 1 homeless person. In Rabai, interviews involved 3 HCPs, 2 administration personnel, 2 teachers, and 2 PLWCDs.

The research and ethics approving body, the National Commission for Science, Technology and Innovations, sent letters to county-level offices informing them about the study. Health facility In-Charges were contacted by the study’s fieldwork team and were informed beforehand. In each area, two local liaisons assisted our research assistants in screening for respondents from their respective areas who met our study’s eligibility criteria. Two youth leaders were also identified to assist in the recruitment of adolescent participants. The recruitment of participants was conducted purposefully by approaching potential participants and obtaining their consent based on inclusion criteria: aged 15 years and above, residing in either Mombasa Old Town or Kaloleni-Rabai, and willing or able to provide assent/consent. Those who agreed were invited to a suitable venue for the IDIs/FGDs. All participants provided consent prior to the interviews. Some participants scheduled for FGDs were replaced if they were unavailable on the designated day.

### 2.3. Fieldwork Team and Data Collection

Data collection and management followed the study protocol and fieldwork guidelines provided during research team training. Three experienced female research assistants (RAs) with health or social science diplomas were competitively recruited. They received two-day training on the study background, research ethics, data collection methods (IDIs and FGDs), study tools, and consent procedures. After the training, the supervisor and RAs collected data for Mombasa and Rabai from 29 March to 1 April 2022.

The RAs used interview guides to conduct the IDIs and FGDs. The data analyzed for this manuscript emanated from two main questions: (1) How do we improve access to COVID-19 vaccination? (2) What are the potential solutions to overcoming barriers to COVID-19 vaccination uptake? The fieldwork was carried out within four working days with two full days per site. The data collection exercise was implemented by the three RAs under the guidance of the fieldwork supervisor. All IDI and FGD sessions were conducted face-to-face and lasted approximately 60 min. The IDIs and FGDs were all captured in notes and audio recordings. The interviews were conducted in person at private venues in health facilities. The FGDs were also held in an in-person setting at health facilities and were led by a trained facilitator using an interview guide that was designed to elicit a discussion on the two domains mentioned above. A trained scribe assisted the main facilitator in warmly welcoming FGD participants to the venue, consenting participants, and note-taking. Consent and assent forms were provided in English and Swahili, and participants signed two copies. One copy was given to the participant, while the study team retained the second copy.

In this study, we address data saturation during the proposal writing stage by engaging a total of 64 participants from two main categories: 18 from the IDIs and 46 FGD participants. This number of data collection sessions was considered adequate to gain the required data because data from IDIs are reported to allow saturation at between 9 and 17 interviews.

### 2.4. Ethical Considerations

This study was guided by a protocol that was approved by the Aga Khan University Institutional Ethics Review Committee, the National Commission for Science Technology and Innovations, Mombasa County, and Kilifi County. All respondents above the age of 18 years provided written consent, and all adolescents below 18 years gave assent, and signed consent was required from their guardians. The study team maintained the confidentiality of all material and information and ensured no identifying information on individual participants was included in any analysis or study reports.

### 2.5. Data Analysis

Following the fieldwork, transcription, and translation of all IDI- and FGD-recorded data, all transcripts were analyzed with the use of NVivo 12 software version 12.6.1. When led by two coders, the study used both deductive and inductive analysis to code all the combined 24 transcripts from 18 IDIs and 6 FGDs. Coders became familiar with transcript data and used these study tools to generate initial codes. As the coding progressed, they added additional ideas emanating from data, thus expanding the code book as necessary. Consequently, identified codes were arranged by site (Kilifi or Rabai), method (IDIs and FGDs), and into thematic categories. Lastly, investigators selected vivid quotes in support of the described themes.

To ensure study rigor and validity, we triangulated the study methods (IDIs and FGDs), sites, and categories of participants. During fieldwork, data analysis, and the interpretation of findings, the research team held ongoing discussion sessions. The research team kept detailed recruitment, consenting, and data records. We also used a well-qualified and trained credible fieldwork and coding team, providing rich and detailed descriptions of the study context, participants, and methodology. The data analysis was clearly and systematically described, and we ensured participants’ voices were represented without the influence of the researchers’ interpretations. See ‘Strengths and limitations of the study’ and ‘Fieldwork Team’ section. Regarding reliability, the IDIs and FGDs were administered by a credible team that was well-trained to consistently run the data collection session using the same tools and maintaining field notes in a systematic way. The coders used a continuous transparent iterative process of applying codes and integrating emerging findings into logical thematic categories in line with study questions. Any discrepancies in coding were discussed by the two analysts to consensus.

## 3. Results

### 3.1. Demographic Characteristics of Study Participants

This study engaged health care providers (HCPs), local administration personnel, teachers, and people living with chronic conditions. Five HCP roles were represented as follows: clinician, nurse, oral health officer, laboratory technician, and public health officer. Their years of service ranged from 2 to 8 years. All HCPs were vaccinated, and one reported having a chronic health condition. There were four local administrative personnel whose roles included Senior Chief, Assistant Chief, and two village elders. The years of serving in their current role ranged from 2 to 10 years. All administrators were vaccinated, with two reporting chronic health conditions. Two of the four teachers interviewed were Madrasa religious teachers, including one secondary school teacher and one nursery schoolteacher. Their years in the role ranged from 4 to 27 years. Only one teacher was vaccinated, and one other had a chronic health condition. In the chronic health condition category, five respondents had conditions that included diabetes mellitus, hypertension, goiter, ulcers, and sickle cell. Two were vaccinated, and one of them was homeless.

The group of 15–19-year-old adolescents had an average age of 16.5 years. All were students; ten were female, and nine were male respondents. Thirteen were in secondary school, while six were in primary school. Only five had been vaccinated, while three had chronic conditions. The 20–49 years group had an average age of 35.5 years. Fifteen were female, while five respondents were male. The majority (eleven) were married, seven were single, and one was divorced. There were eleven self-employed, eight unemployed, and one employed. Half were vaccinated, while only three had chronic health conditions. The 50+ year group had an age range from 50 to 72 years, with an average age of 57.7 years. Nine were female, while eight were male respondents. On marital status, thirteen were married, two were widowed, one was divorced, and one was separated. Only six were vaccinated, and six respondents had chronic health conditions.

The study findings revealed five main themes that captured participants’ ideas on possible ways to reduce barriers to COVID-19 vaccination. These included the following (Table 1): (1) Increase education and awareness on COVID-19; (2) Engage diverse stakeholders; (3) Use diverse materials and methods in the communication and implementation of vaccination programs; (4) Increase vaccination facilities and staff capacity to deliver quality services; (5) Revise relevant government policies and guidelines.

#### 3.1.1. Increased COVID-19 Education and Awareness

The ideas shared under this theme were discussed in rural and urban areas. Firstly, participants believed that inadequate knowledge about the pandemic and COVID-19 vaccines hindered vaccination efforts. All age groups from both sites recommended educating people on COVID-19. Adults from both sites recommended holding outreaches and awareness creation meetings while addressing fears and rumors/misconceptions about COVID-19.

They suggested that by educating people about COVID-19, vaccination rates could increase. Health education and promotion efforts, similar to other routine programs, were considered necessary at health facilities and within the broader community as clients visited for various services.
*We want, when we go to clinics, to be educated, just as they educate us about diet and how to take medication...*(Mombasa, PWCD, IDI 5)
*The only thing that can change this is education… They should be informed that the government cannot bring something that is harmful to their people.*(Rabai, PWCD, IDI 1)

Secondly, participants recommended mobilization and outreach to educate and raise awareness about COVID-19, as well as deliver vaccination services closer to communities. This could be particularly beneficial for those residing in far-flung areas and the economically disadvantaged who struggle to access healthcare facilities regularly. Bringing vaccines closer to these communities could increase the convenience and likelihood of vaccination. Outreaches in institutions and neighborhoods have proven effective in past vaccination programs.
*We have done outreach, and people come from facilities like Aga Khan, Jeffrey’s, and State House. They are going around with cooler boxes, and people are being vaccinated. People who are not able to go to the facilities found in their respective areas are given the vaccines with proper documentation in place.*(Mombasa, Local administrator, IDI 3)
*Vaccination should be conducted similarly to those of children, whereby the healthcare workers visit people in their homes*.(Rabai, FGD 50+>)

A third idea under this theme was to disseminate information to address rumors in communities. Participants recommended dedicated efforts to address fears and misconceptions about COVID-19. The government should find ways to reduce vaccine hesitancy. Increasing an understanding of the pandemic, vaccination locations, and the benefits of vaccination could lead to higher vaccination rates.
*Using real-life examples… show people the benefits of vaccination. Information should be clearly given that though they will get vaccinated, they might still get infected with the virus, but the advantage is that it will not be severe.*(Mombasa, Local administrator, IDI 6)
*People should be exposed to the right information. If they don’t have the right information, they will only have the rumors to believe in.*(Rabai, PWCD, IDI 4)

#### 3.1.2. Engagement of Diverse Stakeholders

Participants at both sites recognized the value of influencers and role models in promoting COVID-19 vaccination. This was recommended by Mombasa adolescents and Rabai adults. Community Health Volunteers (CHVs), Chiefs, village elders, and media influencers were viewed as effective in expanding the reach of health messages. Well-educated influencers could persuade community members to become vaccinated by imparting knowledge about the pandemic, vaccines, and related services.
*Using CHVs, CHWs, Chiefs, community elders, and FM stations to educate people. In FM stations, people can also be given an open forum to discuss specific issues.*(Mombasa, Local administrator, IDI 3)
*We should educate the keynote people in the community: Chiefs, men, and women in the chamas so that they can educate the rest of the people.*(Rabai, HCP, IDI 5)

Rabai adult participants also stressed the importance of teamwork and stakeholder collaboration to reach and educate all communities (urban and rural) about COVID-19 and address misconceptions. Religious leaders were encouraged to equally share available COVID-19 information, which had already been initiated. By working together as a united front, the broader community could benefit. Additionally, relying solely on Assistant Chiefs was inadequate for deep rural areas. Collaborating with various stakeholders, such as organizing education forums, was seen as a promising approach to achieving positive outcomes.

#### 3.1.3. Useful Materials and Methods in Communication and Implementation of COVID-19 Vaccination Programs

At both sites, four ideas were proposed to reduce barriers. The use of testimonies from the vaccinated was recommended by Mombasa adolescents and Rabai adults. Sharing vaccination experiences could inspire others to become vaccinated, especially within their own communities. Vaccinated individuals could serve as ambassadors, challenging negative perceptions and motivating others to receive the vaccine.
*There should also be testimonies from the people who have been vaccinated and are experiencing the benefits of the vaccination.*(Mombasa, Teacher, IDI 7)
*We have people who are already vaccinated… We can use them as testimonies so that people can see that some of the information they have is just rumors.*(Rabai, HCP, IDI 5)

Secondly, participants suggested targeted health messaging for effective sensitization on COVID-19. Adults from both sites recommended targeting different groups/audiences for sensitization, door-to-door sensitization, and the use of incentives. For example, Community Health Volunteers CHVs could engage specific audiences like Boda Boda (motorcycle) riders, adolescents, women, chamas, and community influencers in group discussions. This approach could provide an excellent opportunity to educate them about the pandemic and vaccines.
*For the youth, there are different spots you can get them into. I play football, and after the game, we normally talk. That can also be an opportunity to pass out information regarding vaccines … There is another large population of bodaboda riders. They have their meeting points too. They can be educated in those groups.*(Mombasa, HCP, IDI 4).
*Changing someone’s behavior takes time, so we can still continue doing it, especially focused sensitization meetings targeting particular groups.*(Rabai, HCP, IDI 7).

Moreover, in Mombasa, targeted parents could be sent COVID-19 messages through school-attending lower primary children. In Kilifi, some community members were renowned for alcoholic intake. They were, thus, encouraged to take a break from alcohol in order to become vaccinated. This was yet another unique group to be targeted with COVID-19 education and sensitization.
*I usually tell them that they can sacrifice the alcohol even for just three days so as to enable them to get the vaccine.*(Rabai, Teacher, IDI 8)

Third, a door-to-door approach was deemed beneficial. Prior to home visits, the community could be informed about the presence of health workers in the villages. Village elders and Community Health Volunteers CHVs could communicate that health workers are moving from door to door to sensitize people about vaccination. The health workers would provide accurate information to counteract circulating false information. They would also emphasize the importance of receiving the COVID-19 vaccine.

The fourth idea focused on using incentives to motivate COVID-19 vaccination. One option was offering vaccines that required fewer visits/doses. Providing food and additional health assessments alongside COVID-19 vaccinations could make the visit valuable for the impoverished and unwell community members. These additional assessments would also ensure vaccinations were administered with awareness of any pre-existing conditions.
*If it’s a single dose and you incentivize it, you are sure that a person will get that vaccine once.*(Mombasa, HCP, IDI 1)
*IF doctors and CHVs give food to distribute to the community members during their visits, this will make them more interested in the agenda and, therefore, increase the uptake of the vaccine.*(Rabai, FGD 20–49>)
*The government should seek to provide healthcare services for people suffering from other diseases. They fear that getting the vaccine would make their health condition even worse.*(Rabai, FGD 15–19>)

Additional ideas from Mombasa adults included diversifying methods and mediums for sharing COVID-19 information and vaccination programs. Participants recommended utilizing various mainstream media, social media platforms, video messaging, and written material. The government was encouraged to adapt COVID-19 messages for Facebook, TikTok, Instagram, and WhatsApp to reach those who had not been vaccinated yet. Bringing people together for educational purposes, such as showcasing videos of individuals thriving after vaccination, could create a favorable environment for future voluntary health gatherings.

Mombasa participants suggested targeting places frequented by adolescents to raise awareness among the younger population about the pandemic and available vaccines. They also recommended utilizing collective forums such as schools, mabaraza (heterogenous community gatherings), and political rallies during campaign seasons to disseminate health messages on COVID-19.

In Rabai, an additional idea from adults was the need to focus on promoting positive stories about vaccination and avoiding negative reports. This approach aimed to encourage the use of vaccination services.

#### 3.1.4. Increase Vaccination Facilities and Health Worker Capacity

Three ideas were reported under this theme by adults from both sites. Firstly, adults from both sites recommended increasing the number of vaccination centers to make services more accessible geographically. This could be achieved by utilizing existing health facilities and establishing temporary vaccination centers. Community Health Volunteers (CHVs) could play a crucial role in disseminating information about the expanded vaccination center availability to reach all community members and encourage more people to become vaccinated.

Secondly, adults from both sites thought recruiting and employing additional health workers could enhance COVID-19 vaccination and improve uptake. Having dedicated staff for the vaccination program could increase efficiency and reduce client waiting times.
*If there are a few personnel giving out the vaccine, the coverage will be a little tricky for them. The more personnel there are, the easier it will be to carry out the vaccination process…we can also increase vaccination centers.*(Mombasa, HCP, IDI 4)
*The services can also be improved to have specific staff dealing with vaccines. Sometimes, the staff issuing the vaccines are usually assigned somewhere else, and when clients come, they have to wait to receive the service.*(Rabai, HCP, IDI 6)

Third, adults from both sites recommended increased funding for health promotion, community sensitization, and vaccine provision activities. Adequate funds were necessary to ensure effective logistics for awareness and sensitization efforts. Better funding could also help overcome logistical challenges in educating communities about COVID-19. Once messaging teams reach the communities, vaccine refusal rates are expected to decrease. Moreover, reliable funding and careful attention to supplies and transportation could contribute to a smooth vaccination process by ensuring an adequate stock of materials. Proper transportation, such as well-designed vehicles, could be essential for the timely delivery of vaccines. Addressing the timely availability of quality consumables can ensure clients receive required services and high-quality vaccines, avoiding those close to expiration.
*The vaccine should also be enough. It would not be good if we sensitize people to go for the vaccine only for them to go and find they are not available.*(Rabai, Teacher, IDI 8)
*In this facility, we get our doses from Kilifi. Sometimes, they may say the vaccines are not enough, and we need to share… There was a time it was not that the vaccines were not there, but the ones that were there had expired. If the community members learn that the expiry date of the vaccines is near, they also don’t come for vaccines. There should be a proper calculation of the estimated number of vaccines a community can consume to prevent the wastage of vaccines that could have helped another community.*(Rabai, HCP, IDI 5)

Two additional ideas from Mombasa adult participants addressed the timing and delivery of vaccinations at the facilities. Some individuals were unable to reach the facilities in time before they closed. Participants suggested implementing flexible vaccination schedules to accommodate diverse groups, including youths and adults facing role conflicts. Furthermore, healthcare providers were reminded to conduct thorough examinations before administering vaccines. This would involve conducting a series of medical tests to assess any underlying medical conditions prior to vaccination.

Rabai adult participants had additional ideas under this theme. They believed that providing additional training for all staff involved in the vaccination program could build trust and reduce barriers to COVID-19 vaccination. This training can ensure the proper handling and safe transportation of vaccines, as well as the administration of vaccinations. Users’ positive experiences can be shared within their communities, fostering trust in the vaccination process. They also thought that an aggressive targeted follow-up of the unvaccinated and an improvement of healthcare providers’ attitudes were also seen as effective strategies to increase vaccine uptake. Negative experiences, such as encounters with rude or corrupt staff, could discourage the use of vaccination services, even if other efforts were well-implemented. Therefore, it was crucial to ensure clients received quality service from visiting vaccination centers.
*Doctors should be polite. Someone may be discouraged to visit a vaccination center, even if it’s nearer, if they find out that the health care workers there are rude.*(Rabai, FGD 20–49)
*It is difficult because they [HCPs] are the ones who are supposed to guide us... However, I think the healthcare workers should be human. Some of them were even selling the vaccines.*(Rabai, Teacher, IDI 8)

#### 3.1.5. Government Health Policy and Guidelines

The fifth theme captures thoughts on how the government can help reduce barriers to COVID-19 vaccination by composing policies and guidelines. Adolescent participants from Rabai thought a government policy that made vaccination mandatory would increase utilization.
*The government should make COVID-19 vaccination mandatory. If it is voluntary, most people will never go for the vaccination. Certain restrictions should also be put in place so as to improve the uptake.*(Rabai, FGD, 15–19 years)

Nonetheless, adult participants from Mombasa thought COVID-19 vaccination should remain voluntary. They proposed a focus on educating people about the pandemic and sensitizing them on available vaccinations. This was then likely to result in increased vaccine uptake.
*It should not be mandatory... People should be informed about it first.*(Mombasa, IDI 6)
*To remove the fear of the vaccine, uptake should not be made mandatory. It should be voluntary. When we go to hospitals, we should not be forced or persuaded to take the vaccine.*(Mombasa, IDI 5)

## 4. Discussion

Our results from rural and urban areas of Kenya indicate several methods to tackle vaccine hesitancy and boost vaccine uptake, the first being to level up awareness campaigns by delivering high-quality information on COVID-19 and its vaccines. This was also shown in the literature as a recent study in Kenya found that we should use customized messages to reach people with different concerns and different levels of education [15]; furthermore, accurate and transparent information from trusted sources is essential to increase vaccine rollout [15], which was also the case in Ethiopia where a study showed that health education and communication from governmental authorities are very crucial methods to alleviate the negative attitude of the COVID-19 vaccine [16]. Another study from Pakistan indicated that the main reason for the low rate of vaccine acceptance was a lack of knowledge, understanding, and perception of the risks and safety of the COVID-19 vaccine; hence, strategies to raise awareness of the benefits of vaccination should mainly target individuals in lower socioeconomic groups [17].

We believe that if awareness campaigns address people’s questions and concerns effectively, the majority of information gaps regarding the COVID-19 vaccine can be filled. Consequently, there will be fewer misconceptions and rumors to debunk. Nonetheless, due to the digital, hyper-connected society such as the present one, misinformation is likely to accompany every health crisis [18]. Our respondents have also described the problem of infodemics as one of the main barriers toward vaccination and wished for more information transparency and efforts to debunk rumors around the COVID-19 vaccine. Evidence from the literature shows that debunking misinformation has a limited role and is insufficient once rumors have circulated [19,20]. However, other studies have indicated that using debunking in media campaigns on top of vaccine information and social norm modeling could be an adequate technique to tackle misinformation [21], especially when engaging the community and using vaccine champions. This strong community engagement is a useful approach when managing vaccine hesitancy alongside communication campaigns and evidence-based interpersonal communication [22].

Our respondents have similarly indicated the importance of community engagement, the use of influencers and role models, and encouraging stakeholders to work together, which is also supported by the literature as a study in thirteen African countries has emphasized the importance of African countries to address the challenges facing RCCE (risk communication and community engagement) to contain the pandemic effectively and to prepare for future public health emergencies [23]. As understood from the debunking method mentioned above, no single strategy can fit all to boost vaccine acceptance. Therefore, our respondents have indicated that using various techniques and testing what works and is suitable for our specific community is key to reducing barriers to vaccination uptake in Kenya. Moreover, evidence from the literature also supports these claims as a review of published reviews on vaccine hesitancy concluded that it is crucial to understand the specific concerns of the various groups of vaccine-hesitant individuals because a compelling “one size fits all” intervention is unlikely ever to exist [24].

One of the prominent solutions that our respondent suggested for the context of Kenya is sensitization (raising awareness and educating people about the importance and safety of vaccines to encourage more vaccinations), whether for groups of the community during gatherings, baraza gatherings, political campaigns, or even individually, via a door-to-door approach. A baraza is a traditional community assembly that is common in East African culture and has been used as a novel methodology in community-based participatory health research [25,26]. Past studies also favored the sensitization method, such as a Cameroonian study during COVID-19 that highlighted the role of good communication in addressing vaccine hesitancy [27]. Another study in northern Nigeria stated that incorporating door-to-door visits with media campaigns may increase childhood immunization uptake [28]. Another debatable strategy our informants mentioned, in combination with the other measures to boost vaccine acceptance, is the use of financial incentives. A financial incentive method was highly criticized in the literature as several studies yielded that small financial incentives do not meaningfully improve COVID-19 vaccination rates among the vaccine-hesitant [29,30]. However, other studies indicated that financial incentives increased vaccination rates among adults when combined with other measures and multicomponent and dialogue-based interventions [8,31], which our informants also suggested.

Our interviewees believe that to prepare for the next pandemic or during health emergencies, the government must efficiently communicate timely, accurate health information on social media platforms and video messaging; studies from the literature also emphasize that governments and health authorities should consider providing health information and the use of films on social media as an affordable yet compelling and effective tool for boosting willingness to vaccinate against COVID-19 among populations most reluctant to receive them [32,33].

Capacity building in terms of increasing vaccination centers and trained staff has played a considerable role in our results. A study conducted on the continent of Africa emphasized the role of capacity building as it is essential to enhance and sustain public demand for vaccination and maintain the remarkable accomplishments of vaccination programs on the continent. Increasing the number of vaccine centers also means addressing the problem of inequity and reaching out to the most vulnerable and out-of-reach populations [34]; these marginalized populations cannot leave their jobs to become vaccinated, and even if they did, they would reach the closest vaccine center, when it is already closed. Results from another study on the influenza vaccine have indicated the essential role of increasing the number of training courses for vaccine staff, as this was shown to boost influenza vaccine uptake [35].

The last theme in our results, which was supported by interviewees from rural areas but was not supported by interviewees from urban areas, is making vaccination mandatory. Some results from the literature are in favor of compulsory vaccination, like the one conducted in Europe, which justified their conclusion by taking the measles example, which was associated with lower measles incidence for countries with mandatory vaccination without nonmedical exemptions [36]. Another study also favored the mandatory vaccination of healthcare workers HCWs in the case of influenza, which was introduced in the United States a decade ago with outstanding outcomes [37]. However, a different study concluded that herd immunity (where a substantial portion of the population is immune to the disease either through vaccination or prior infection) could be reached without a policy of compulsory vaccination, and none the less the same study stated that mandatory policy might be found admissible, too, were it to become critical [38]. Some interesting results were also found from other forms of research with only health workers in mind; this particular study has yielded that if voluntary vaccine uptake fails to achieve the desired rates, mandatory policies should be considered; this would be applicable only if the benefits outweighed the harm for HCWs, if patient welfare was enhanced, and fair rules and exemptions defined [39].

Mandatory training for doctors in patient care was mentioned in our IDI and FGD, which we think is crucial to ensure patients are treated with the utmost respect, avoiding any form of rudeness or impoliteness. Such training should encompass effective communication skills, cultural sensitivity, empathy, and active listening. Teaching conflict resolution and stress management techniques can also be beneficial, helping doctors maintain composure and professionalism during challenging interactions in future pandemics. By emphasizing these skills through ongoing training, the health system in Kenya can enhance the doctor–patient relationship, fostering trust and collaboration and ultimately boosting vaccination uptake. Furthermore, in Kenya, healthcare workers involved in selling vaccines should face disciplinary measures like suspension or termination, along with legal actions such as fines and potential imprisonment. Public awareness campaigns are crucial to discourage this illegal and unethical practice.

### Strengths and Limitations of the Study

The strength of this qualitative study resides in its methodological rigor, characterized by the utilization of complementary multi-qualitative methods, specifically focus group discussions (FGDs) and in-depth interviews (IDIs), to capture a diverse range of perspectives among both adult and adolescent participants. Moreover, this study encompassed two distinct settings, namely Mombasa Old Town, representing an urban environment, and Rabai, Kilifi, representing a rural context. This deliberate inclusion of participants from different settings enhances this study’s validity and enriches an understanding of the research phenomenon under investigation.

Despite the mentioned strengths, this study also encountered some limitations. First, the findings may lack generalizability due to the specific contexts of Kenya and the sample size involved. Caution should be exercised when applying the results to populations of different settings. Second, although our research team conducted the necessary advanced training and experiences, the potential for researcher subjectivity and bias may have influenced the data collection, analysis, or interpretation processes. Hence, we implemented measures like peer reviews, detailed note-taking, and transparent reporting to reduce the likelihood of biases. Acknowledging these limitations is crucial for a comprehensive understanding of the study’s findings and to inform future research efforts.

## 5. Conclusions

In this study, we engaged diverse Kenyan stakeholders to explore ideas that could potentially be applied to reduce barriers to COVID-19 vaccination uptake. Awareness campaigns should prioritize filling potential information gaps with high-quality information before rumors emerge rather than relying on the debunking method after rumors evolve into narratives. Engaging diverse stakeholders and strong community engagement via influencers and vaccine champions on top of the debunking method is key to restoring routine immunization. Additionally, employing various communication techniques in combination with previously mentioned strategies and testing what works and is suitable for specific communities, e.g., sensitization, incentives, and video messaging on social media, could significantly increase vaccine uptake. Furthermore, increasing vaccination centers and trained staff is highly valuable when addressing the problem of inequities and, hence, increasing vaccine uptake. In the event of voluntary vaccine uptake, when failing to attain the desired rates, careful consideration should be given to the revision of pertinent government health policies and guidelines, with the potential incorporation of mandatory policies. However, such a course of action should only be pursued if a comprehensive assessment indicates that the benefits of implementing mandatory vaccination outweigh any associated drawbacks or potential harm.

## Figures and Tables

**Table 1 ijerph-20-07093-t001:** How to reduce barriers to COVID-19 vaccination and improve access in Mombasa and Kilifi Kenya.

Themes	Specific Idea/Code	Mombasa Old Town (Urban)		Rabai (Rural)	
		Adolescents	Adults	Adolescents	Adults
1. Increase education and awareness of COVID-19	Educate people on COVID-19	X	X	X	X
	Hold outreaches and awareness creation meetings		X		X
	Address fears, rumors/misconceptions about COVID-19		X		X
2. Engage diverse stakeholders	Use influencers and role models	X			X
	Encourage stakeholders to work together				X
3. Use diverse materials and methods in the communication and implementation of vaccination programs	Use testimonies from the vaccinated	X			X
	Target different groups/audiences for sensitization		X		X
	Door-to-door sensitization		X		X
	Use of incentives		X		X
	Use diverse social media platforms		X		
	Use written educational materials		X		
	Use video messaging		X		
	Targeting places frequented by adolescents		X		
	Use of baraza (heterogeneous community gathering)		X		
	Use political rally platforms		X		
	Trend positive vaccine information				X
4. Increase vaccination facilities and staff capacity to deliver quality services	Increase vaccination centers/closer to people		X		X
	Increase the number of staff		X		X
	Fund awareness logistics and reduce stock out of materials		X		X
	Have time flexibility on vaccination services		X		
	Proper health assessment before issuing vaccines		X		
	Proper handling of vaccines				X
	Do follow-up of the unvaccinated				X
	Expand training of staff on the vaccination program				X
	Address providers’ poor attitudes				X
5. Revise relevant government health policies and guidelines	Make COVID-19 vaccination mandatory			X	

## Data Availability

Data from this study cannot be shared publicly due to ethical considerations when handling transcribed data. Researchers interested in study data can contact the corresponding author Ahmed Asa’ad Al-Aghbari1 [ahmedalaghbari24@gmail.com] and AKUKenya.ResearchOffice@aku.edu.

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
