# Peer review of "Reducing Barriers to COVID-19 Vaccination Uptake: Community Ideas from Urban and Rural Kenya"

_ijerph, 2023, doi:10.3390/ijerph20237093_

Round 1
Reviewer 1 Report
Comments and Suggestions for Authors
Author Response
Dear Editor,
Manuscript title: Reducing Barriers to COVID-19 Vaccination Uptake: Community Ideas from Urban and Rural Kenya
Manuscript ID: ijerph-2588268
We would like to extend our heartfelt thanks for your helpful review of our article. Your edits and recommendations have greatly enhanced our work. Below, we respond to each of the comments received.
Response to review
Reviewer 1
Thanks for your effort to shed light on methods to tackle vaccine hesitancy in Kenya as a LMIC. My comments are as follows.
Response: Thank you for your review and commendation.
Introduction
- I understand that the authors focus on vaccine hesitancy as a barrier to vaccine uptake. But based on the title of the manuscript, I believe that barriers should be explained and talked about more.
Response: A new paragraph about vaccine barriers was added. Page 2 line (82-87)
- Line 67, LMIC: please spell out the full term at its first mention.
Response: The full term of the abbreviation LMCs was written in full. Page 2 line (62)
- Line 107-129, It is not well fitted in the introduction. The relation between infodemic and the introduction is not clear for me. I prefer to delete this paragraph.
Response: The paragraph was deleted.
- Please provide some data about vaccination uptake barriers in other countries.
Response: A new paragraph about vaccination uptake barriers in other countries was added. Page 3 line (98-104)
Methods
- The subsection "demographic characteristics of study participants" should move down to the results section.
Response: The subsection "demographic characteristics of study participants" was moved to the results section as suggested. Page 5 line (192-219)
- Please provide more details about FGDs and how they were conducted.
Response: A new paragraph about FGDs and how they were conducted was added. Page4 line (150-155)
- Please provide details about rigor, validity, and reliability of the qualitative method you administer.
Response: This information was captured in the methodology sections. We however have added a paragraph specifically highlighting study rigor, validity, and reliability.
Page 4, line (178-189).
- Please provide more details about data analysis and about how you merged the results of FGDs and IDls.
Response: More details about data analysis have been added. Page 4, line (178-189).
- Did you consider data saturation?
Response: Yes we intentionally considered data saturation during the study design stage. We have added information about data saturation. Page 4 lines (156-159)
Results
Is it possible to bring results considering age groups too.
Response: We have added findings by site and age (adolescents vs. adult participants). We have also updated Table 1 accordingly. Page 5,6 line (192-219)
Discussion
Line 494: edit this: "RCCE risk communication and community engagement"
Response: The abbreviation was written fully. Page11 line (459)
Thank you,
AHMED ASA’AD AL-AGHBARI (On behalf of all authors)

Reviewer 2 Report
Comments and Suggestions for Authors
thank you for submitting this interesting qualitative study exploring ways to overcome barriers to COVID-19 vaccination in urban & rural Kenya; the participants are appropriately diverse (age, rural vs urban etc) and the results you present are interesting and should help inform vaccine policy/ uptake in Kenya.
at times, the manuscript would benefit from being more succinct (particularly Introduction and Discussion) and, in some cases, the message would be made clearer by using plain English (simpler more straightforward terms).
ABSTRACT
Line 24 - 25; suggest re-write opening sentence to better indicate purpose of this study.
Line 35 - suggest 'barriers to COVID vaccination' (not COVID vaccine)
INTRODUCTION
- general comment; could be shortened/ more succinct
- Line 47 ; pleaes note 'coronovirus disease 2019' not routinely used in English, fine to use COVID-19 (or SARS-CoV2)
Line 54 - 55; suggest re-write 'revving the ending'
Line 59; please clarify daily vaccine administration total - when was this measured? it this figure still current in 2023?
Line 63; please clarify 'initial protocol ' in Kenya - how many doses?
Line 74; suggest re-write 'various-factor event coaxed by a scope of factors'
Line 76; reconsider 'deviates'
Line 84; reconsider 'delimited'
Line 89; reconsider 'arise with strategies...'
Line 92 ' reconsider 'delimited'
Line 112; reconsider 'others have transpired...'
Line 114 reconsider 'risen use'
Line 131 - suggest 'to support the rollout' (rather than subsidise)
Line 132 suggest remove 'better'
METHODS
Line 161; more details about how researchers identified potential participants to approach
Line 208: consider presenting demographic details in results section & possibly summarise this information in a table
RESULTS
Line 242 - remove 'on' or 'of'
Line 330 - please clarify advice re alcohol; abstaining not part of international COVID-19 vaccination recommendations
DISCUSSION
- could be more succinct
Line 461 - consider re-writing ' tailored messages should be needed to attain those with different concerns ..'
Line 478 remove 'bound'
Line 507 the term 'sensitization' is used throughout the manuscript; consider defining exactly what this term means in this context
Line 514 consider re-write 'brave the likelihood of hesitancy
Line 547 consider re-write 'level up the influenza vaccine uptake'
CONCLUSION
- no comments
Comments on the Quality of English Language
there is room to be more succinct - especially Introduction and Discussion
at times - the readability of the manuscript could be improved by using plain English/ simpler terms (several examples identified above)
Author Response
Thank you for submitting this interesting qualitative study exploring ways to overcome barriers to COVID-19 vaccination in urban & rural Kenya; the participants are appropriately diverse (age, rural vs urban etc) and the results you present are interesting and should help inform vaccine policy/ uptake in Kenya.
Response: Thank you for your review and commendation.
At times, the manuscript would benefit from being more succinct (particularly Introduction and
Discussion) and, in some cases, the message would be made clearer by using plain English (simpler more straightforward terms).
Response: We have made the manuscript more succinct and used clearer plain English language.
ABSTRACT
Line 24 - 25; suggest re-write opening sentence to better indicate purpose of this study.
Response: The opening statement was rewritten to better indicate the purpose of the study. Page1, Line (24-26)
Line 35 - suggest & barriers to COVID vaccination (not COVID vaccine)
Response: The term vaccination' was used instead of vaccine. Page 1, Line (34)
INTRODUCTION
- general comment; could be shortened/ more succinct
Response: We have shortened the introduction. A paragraph about infodemic was deleted.
- Line 47 ; please note 'coronovirus disease 2019' not routinely used in English, fine to use COVID-19 (or SARS-CoV2)
Response: We have edited accordingly and used ‘COVID-19’ throughout the manuscript.
Line 54 - 55; suggest re-write 'revving the ending'
Response: We have edited to ‘accelerating the end of the pandemic’. Page 2, Line (53)
Line 59; please clarify daily vaccine administration total - when was this measured? it this figure still current in 2023?
Response: We have updated the information to August 2023. Page 2, Line (54-59)
Line 63; please clarify 'initial protocol ' in Kenya - how many doses?
Response: We have updated the information to ‘two doses as full vaccination.’
Page 2, Line (59)
Line 74; suggest re-write 'various-factor event coaxed by a scope of factors'
Response: We have edited to ‘multifaceted phenomenon influenced by a range of factors’
Page 2 line (67-68)
Line 76; reconsider 'deviates'
Response: We have edited to ‘varies’. Page 2, Line (69)
Line 84; reconsider 'delimited'
Response: We have edited to ‘constrained’. Page 2, Line (74)
Line 89; reconsider 'arise with strategies...'
Response: We have edited to ‘develop strategies to manage vaccine hesitancy’ .
Page 2, Line (78)
Line 92 ' reconsider 'delimited'
Response: We have edited to ‘constrained resources’ Page 2, Line 80
Line 112; reconsider 'others have transpired...'
Response: We have deleted the whole paragraph as advised by a reviewer.
Line 114 reconsider 'risen use'
Response: We have deleted the whole paragraph as advised by a reviewer.
Line 131 - suggest 'to support the rollout' (rather than subsidise)
Response: Thank you. We have deleted this paragraph as advised by a reviewer.
Line 132 suggest remove 'better'
Response: Thank you. We have removed ‘better’ Page 3, Line 106
METHODS
Line 161; more details about how researchers identified potential participants to approach
Response: Details have been added. Page 3, Line (127-130)
Line 208: consider presenting demographic details in results section & possibly summarise this information in a table
Response: The demographic subsection was moved to the results section.
Page 5 Line (192-219)
RESULTS
Line 242 - remove 'on' or 'of'
Response: the proposition “of” was deleted. Page 7, Line (238)
Line 330 - please clarify advice re alcohol; abstaining not part of international COVID-19 vaccination recommendations
Response: A clarification was added. Page 8, Line (312-313)
DISCUSSION
- could be more succinct
Response: We have made the discussion section more succinct
Line 461 - consider re-writing ' tailored messages should be needed to attain those with different concerns ..'
Response: We have edited to “we should use customized messages to reach people with different concerns and different levels of education” Page 26, Line (434-435)
Line 478 remove 'bound'
Response: We have removed ‘bound’. Page 11, Line (446)
Line 507 the term 'sensitization' is used throughout the manuscript; consider defining exactly what this term means in this context
Response: A clarification was added. Page 11, Line (469-470)
Line 514 consider re-write 'brave the likelihood of hesitancy
Response: We have edited to “Past studies also favored the sensitization method such as a Cameroonian study during COVID-19 that highlighted the role of good communication in addressing vaccine hesitancy” Page 11, Line (473-475)
Line 547 consider re-write 'level up the influenza vaccine uptake'
Response: We have edited to “boost the influenza vaccine uptake” Page 12, Line 498
Thank you,
AHMED ASA’AD AL-AGHBARI (On behalf of all authors)

Reviewer 3 Report
Comments and Suggestions for Authors
This study investigated ideas on ways to reduce barriers to COVID-19 vaccination uptake. It explores methods to overcome COVID-19 vaccine barriers through qualitative research: interviews and group discussions involving healthcare providers, administration personnel, teachers, and individuals with chronic conditions across urban (Mombasa) and rural (Kilifi) Kenya. Audio-recorded discussions were transcribed and thematically analyzed across locations. Five themes emerged in our results regarding how to reduce barriers to the COVID vaccine in the context of Kenya, including awareness campaigns, engaging diverse stakeholders, using various communication techniques, capacity building to increase vaccination centers and trained staff, and lastly, revising relevant government health policies and guidelines. These results indicate the importance of adopting multiple approaches, as no single strategy could boost vaccine acceptance.
The authors provided a work that was informative, extremely well written, and provides a great foundation for future research. The methodological rigor was impressive and map and numerous figures in the manuscript were very detailed and provided excellent clarity. I enjoyed reading this manuscript, and recommend the authors carefully read the manuscript and correct any spelling and/or grammatical errors. It is in the spirit of improving the manuscript that I offer the following minor recommendations:
On Page 2, you wrote:
The enormous inequality in healthcare resources, particularly during a pandemic, undoubtedly worsens the already increasing mistrust in healthcare systems, experts, and the government, which could promote vaccine hesitancy among the LMICs population [3]. Vaccine inequity remains an onerous problem globally, predominantly in LMICs.
· What is LMIC? Spell it out and put (LMIC) behind it.
On Page 3, you wrote:
The WHO defines an infodemic as "An infodemic is an acute outpouring of information, including potentially misleading or inaccurate information that, in a digital, hyper-connected society such as the present one, is likely bound to accompany every epidemic or acute health crisis." [14].
Change to:
The WHO defines an infodemic as "an acute outpouring of information, including potentially misleading or inaccurate information that, in a digital, hyper-connected society such as the present one, is likely bound to accompany every epidemic or acute health crisis." [14].
On Page 5 you wrote:
This study engaged health care providers (HCPs), local administration personnel, teachers, and people living with chronic conditions. Five HCP roles were represented: clinician, nurse, oral health officer, laboratory technician, and public health officer. Their years of service ranged from 2-8 years. All HCPs were vaccinated, and one has reported having a chronic health condition. There were four local administrative personnel whose roles included senior chief, assistant chief, and two village elders. Their years of serving in the current role ranged from 2 -10 years. All administrators were vaccinated, with two reporting chronic health conditions. Two of the four teachers interviewed were Madrasa religious teachers, one secondary school teacher, and one nursery school teacher. Their years in the role ranged from 4- 27 years. Only one teacher was vaccinated, and one other had a chronic health condition. In the chronic health condition category, the five respondents had conditions that included diabetes mellitus, hypertension, goiter, ulcers, and sickle cell. Two were vaccinated, and one of them was homeless.
· What was the age range and average age of the health care providers?
On Page 7 you wrote:
Using CHVs, CHWs, chiefs, community elders, and FM stations to educate people.
· Describe what a CHV and a CHW is. Spell out the name and put (CHVs) or (CHWs) behind it
On Page 8 you wrote:
For example, Community Health Valunteers CHVs could engage specific audiences like Boda Boda (motocycle) riders, adolescents, women, chamas, and community influencers in group discussions.
It was not until this page that I understood what a CHV is. Provide this definition and others on the previous page (Page 7):
Also, change to:
For example, Community Health Volunteers CHVs could engage specific audiences like Boda Boda (motocycle) riders, adolescents, women, chamas, and community influencers in group discussions.
DISCUSSION
On Page 11 you wrote:
Our respondents have similarly indicated the importance of community engagement, the use of influencers and role models, and encouraging stakeholders to work together, which is also supported by the literature as a study in thirteen African countries have emphasized the importance for African countries to address the challenges facing RCCE risk communication and community engagement to contain the pandemic effectively and to prepare for future public health emergencies [23]
Change to:
Our respondents have similarly indicated the importance of community engagement, the use of influencers and role models, and encouraging stakeholders to work together, which is also supported by the literature as a study in thirteen African countries have emphasized the importance for African countries to address the challenges facing RCCE (risk communication and community engagement) to contain the pandemic effectively and to prepare for future public health emergencies [23]
On Page 12, you wrote:
The financial incentive method was highly criticized in the literature as several studies yielded that small financial incentives do not meaningfully improve COVID-19 vaccination rates among vaccine-hesitant [29,30].
Change to:
The financial incentive method was highly criticized in the literature as several studies yielded that small financial incentives do not meaningfully improve COVID-19 vaccination rates among the vaccine-hesitant [29,30].
On Page 12 you wrote:
those marginalized populations can't leave their job to get vaccinated, and even if they did, they reach the closest vaccine center, when it is already closed.
Change to:
those marginalized populations cannot leave their job to get vaccinated, and even if they did, they reach the closest vaccine center, when it is already closed.
· Do not use contractions in academic writing (unless it is a direct quote)
On Page 12 you wrote:
Results from another study on the influenza vaccine have indicated the essential role of increasing the number of training for the vaccine staff as this had shown to level up the influenza vaccine uptake [36].
Change to:
Results from another study on the influenza vaccine have indicated the essential role of increasing the number of trainings for vaccine staff as this was shown to level up the influenza vaccine uptake [36].
On Page 12 you wrote:
The last theme in our results, which was supported by interviewees from rural areas but refused by interviewees from urban areas, is making vaccination mandatory.
Change to:
The last theme in our results, which was supported by interviewees from rural areas but was not supported by interviewees from urban areas, is making the vaccination mandatory.
On Page 12 you wrote:
However, a different study concluded that herd immunity could be 558 reached without a policy of compulsory vaccination, none the less the 559 same study stated that mandatory policy might be found admissible, too, 560 were it to become critical [39].
· Define what you mean by “herd immunity”
On Page 13, you wrote:
this should be applicable only if benefits outweigh harm for HCWs, patients' welfare is enhanced, and fair rules and exemptions are defined [40].
Change to:
this should be applicable only if the benefits outweigh harm for HCWs, patients' welfare is enhanced, and fair rules and exemptions are defined [40].
On Page 13 you wrote:
Second, although our researcher has conducted the necessary advanced trainings and experiences, the potential for researcher subjectivity and bias may have influenced the data collection, analysis, and interpretation processes.
· How did the researcher decrease bias?
Other Issues:
· What is the theoretical framework on which your study is built? A theoretical foundation would make your work stronger.
On Page 11, you wrote:
Doctors should be polite. Someone may be discouraged to visit a vaccination centre even if it's nearer if they find out that the health care workers there are rude. (Rabai, FGD 20-49) It is difficult because they [HCPs] are the ones who are supposed to guide us... However, I think the healthcare workers should be human. Some of them were even selling the vaccines. (Rabai, Teacher, IDI 8)
· Doctors need to treat patients with respect. They should not be rude (impolite). Discuss mandatory trainings that doctors should receive related to patient care.
· Selling of vaccines is illegal and unethical. Discuss the specific actions that should be taken against healthcare workers that engage in this illegal and unethical practice.
While I found your table to be very succinct, I believe that adding a Conceptual Framework for your work would bring added depth.
Comments on the Quality of English Language
This study investigated ideas on ways to reduce barriers to COVID-19 vaccination uptake. It explores methods to overcome COVID-19 vaccine barriers through qualitative research: interviews and group discussions involving healthcare providers, administration personnel, teachers, and individuals with chronic conditions across urban (Mombasa) and rural (Kilifi) Kenya. Audio-recorded discussions were transcribed and thematically analyzed across locations. Five themes emerged in our results regarding how to reduce barriers to the COVID vaccine in the context of Kenya, including awareness campaigns, engaging diverse stakeholders, using various communication techniques, capacity building to increase vaccination centers and trained staff, and lastly, revising relevant government health policies and guidelines. These results indicate the importance of adopting multiple approaches, as no single strategy could boost vaccine acceptance.
The authors provided a work that was informative, extremely well written, and provides a great foundation for future research. The methodological rigor was impressive and map and numerous figures in the manuscript were very detailed and provided excellent clarity. I enjoyed reading this manuscript, and recommend the authors carefully read the manuscript and correct any spelling and/or grammatical errors. It is in the spirit of improving the manuscript that I offer the following minor recommendations:
On Page 2, you wrote:
The enormous inequality in healthcare resources, particularly during a pandemic, undoubtedly worsens the already increasing mistrust in healthcare systems, experts, and the government, which could promote vaccine hesitancy among the LMICs population [3]. Vaccine inequity remains an onerous problem globally, predominantly in LMICs.
· What is LMIC? Spell it out and put (LMIC) behind it.
On Page 3, you wrote:
The WHO defines an infodemic as "An infodemic is an acute outpouring of information, including potentially misleading or inaccurate information that, in a digital, hyper-connected society such as the present one, is likely bound to accompany every epidemic or acute health crisis." [14].
Change to:
The WHO defines an infodemic as "an acute outpouring of information, including potentially misleading or inaccurate information that, in a digital, hyper-connected society such as the present one, is likely bound to accompany every epidemic or acute health crisis." [14].
On Page 5 you wrote:
This study engaged health care providers (HCPs), local administration personnel, teachers, and people living with chronic conditions. Five HCP roles were represented: clinician, nurse, oral health officer, laboratory technician, and public health officer. Their years of service ranged from 2-8 years. All HCPs were vaccinated, and one has reported having a chronic health condition. There were four local administrative personnel whose roles included senior chief, assistant chief, and two village elders. Their years of serving in the current role ranged from 2 -10 years. All administrators were vaccinated, with two reporting chronic health conditions. Two of the four teachers interviewed were Madrasa religious teachers, one secondary school teacher, and one nursery school teacher. Their years in the role ranged from 4- 27 years. Only one teacher was vaccinated, and one other had a chronic health condition. In the chronic health condition category, the five respondents had conditions that included diabetes mellitus, hypertension, goiter, ulcers, and sickle cell. Two were vaccinated, and one of them was homeless.
· What was the age range and average age of the health care providers?
On Page 7 you wrote:
Using CHVs, CHWs, chiefs, community elders, and FM stations to educate people.
· Describe what a CHV and a CHW is. Spell out the name and put (CHVs) or (CHWs) behind it
On Page 8 you wrote:
For example, Community Health Valunteers CHVs could engage specific audiences like Boda Boda (motocycle) riders, adolescents, women, chamas, and community influencers in group discussions.
It was not until this page that I understood what a CHV is. Provide this definition and others on the previous page (Page 7):
Also, change to:
For example, Community Health Volunteers CHVs could engage specific audiences like Boda Boda (motocycle) riders, adolescents, women, chamas, and community influencers in group discussions.
DISCUSSION
On Page 11 you wrote:
Our respondents have similarly indicated the importance of community engagement, the use of influencers and role models, and encouraging stakeholders to work together, which is also supported by the literature as a study in thirteen African countries have emphasized the importance for African countries to address the challenges facing RCCE risk communication and community engagement to contain the pandemic effectively and to prepare for future public health emergencies [23]
Change to:
Our respondents have similarly indicated the importance of community engagement, the use of influencers and role models, and encouraging stakeholders to work together, which is also supported by the literature as a study in thirteen African countries have emphasized the importance for African countries to address the challenges facing RCCE (risk communication and community engagement) to contain the pandemic effectively and to prepare for future public health emergencies [23]
On Page 12, you wrote:
The financial incentive method was highly criticized in the literature as several studies yielded that small financial incentives do not meaningfully improve COVID-19 vaccination rates among vaccine-hesitant [29,30].
Change to:
The financial incentive method was highly criticized in the literature as several studies yielded that small financial incentives do not meaningfully improve COVID-19 vaccination rates among the vaccine-hesitant [29,30].
On Page 12 you wrote:
those marginalized populations can't leave their job to get vaccinated, and even if they did, they reach the closest vaccine center, when it is already closed.
Change to:
those marginalized populations cannot leave their job to get vaccinated, and even if they did, they reach the closest vaccine center, when it is already closed.
· Do not use contractions in academic writing (unless it is a direct quote)
On Page 12 you wrote:
Results from another study on the influenza vaccine have indicated the essential role of increasing the number of training for the vaccine staff as this had shown to level up the influenza vaccine uptake [36].
Change to:
Results from another study on the influenza vaccine have indicated the essential role of increasing the number of trainings for vaccine staff as this was shown to level up the influenza vaccine uptake [36].
On Page 12 you wrote:
The last theme in our results, which was supported by interviewees from rural areas but refused by interviewees from urban areas, is making vaccination mandatory.
Change to:
The last theme in our results, which was supported by interviewees from rural areas but was not supported by interviewees from urban areas, is making the vaccination mandatory.
On Page 12 you wrote:
However, a different study concluded that herd immunity could be 558 reached without a policy of compulsory vaccination, none the less the 559 same study stated that mandatory policy might be found admissible, too, 560 were it to become critical [39].
· Define what you mean by “herd immunity”
On Page 13, you wrote:
this should be applicable only if benefits outweigh harm for HCWs, patients' welfare is enhanced, and fair rules and exemptions are defined [40].
Change to:
this should be applicable only if the benefits outweigh harm for HCWs, patients' welfare is enhanced, and fair rules and exemptions are defined [40].
On Page 13 you wrote:
Second, although our researcher has conducted the necessary advanced trainings and experiences, the potential for researcher subjectivity and bias may have influenced the data collection, analysis, and interpretation processes.
· How did the researcher decrease bias?
Other Issues:
· What is the theoretical framework on which your study is built? A theoretical foundation would make your work stronger.
On Page 11, you wrote:
Doctors should be polite. Someone may be discouraged to visit a vaccination centre even if it's nearer if they find out that the health care workers there are rude. (Rabai, FGD 20-49) It is difficult because they [HCPs] are the ones who are supposed to guide us... However, I think the healthcare workers should be human. Some of them were even selling the vaccines. (Rabai, Teacher, IDI 8)
· Doctors need to treat patients with respect. They should not be rude (impolite). Discuss mandatory trainings that doctors should receive related to patient care.
· Selling of vaccines is illegal and unethical. Discuss the specific actions that should be taken against healthcare workers that engage in this illegal and unethical practice.
While I found your table to be very succinct, I believe that adding a Conceptual Framework for your work would bring added depth.
Author Response
The authors provided a work that was informative, extremely well written, and provides a great foundation for future research. The methodological rigor was impressive and map and numerous figures in the manuscript were very detailed and provided excellent clarity. I enjoyed reading this manuscript, and recommend the authors carefully read the manuscript and correct any spelling and/or grammatical errors. It is in the spirit of improving the manuscript that I offer the following minor recommendations.
Response: Thank you for your review and commendation. We have corrected spelling and/or grammatical errors. We have also addressed the specific edits that you suggested.
On Page 2, you wrote: The enormous inequality in healthcare resources, particularly during a pandemic, undoubtedly worsens the already increasing mistrust in healthcare systems, experts, and the government, which could promote vaccine hesitancy among the LMICs population [3]. Vaccine inequity remains an onerous problem globally, predominantly in LMICs.
- What is LMIC? Spell it out and put (LMIC) behind it.
Response: The whole abbreviation was spelt out throughout the manuscript. Page 2, Line 62
On Page 3, you wrote: The WHO defines an infodemic as "An infodemic is an acute outpouring of information, including potentially misleading or inaccurate information that, in a digital, hyper-connected society such as the present one, is likely bound to accompany every epidemic or acute health crisis." [14].
Change to: The WHO defines an infodemic as "an acute outpouring of information, including potentially misleading or inaccurate information that, in a digital, hyper-connected society such as the present one, is likely bound to accompany every epidemic or acute health crisis." [14].
Response: Thank you. We have deleted this paragraph as advised.
On Page 5 you wrote: This study engaged health care providers (HCPs), local administration personnel, teachers, and people living with chronic conditions. Five HCP roles were represented: clinician, nurse, oral health officer, laboratory technician, and public health officer. Their years of service ranged from 2-8 years. All HCPs were vaccinated, and one has reported having a chronic health condition. There were four local administrative personnel whose roles included senior chief, assistant chief, and two village elders. Their years of serving in the current role ranged from 2 -10 years. All administrators were vaccinated, with two reporting chronic health conditions. Two of the four teachers interviewed were Madrasa religious teachers, one secondary school teacher, and one nursery school teacher. Their years in the role ranged from 4- 27 years. Only one teacher was vaccinated, and one other had a chronic health condition. In the chronic health condition category, the five respondents had conditions that included diabetes mellitus, hypertension, goiter, ulcers, and sickle cell. Two were vaccinated, and one of them was homeless.
- What was the age range and average age of the health care providers?
Response: All IDIs (including with HCPs) were held with adults (>18 years of age), and we did not collect personal data on number of years lived. This variable was not deemed necessary in this study.
On Page 7 you wrote: Using CHVs, CHWs, chiefs, community elders, and FM stations to educate people.
- Describe what a CHV and a CHW is. Spell out the name and put (CHVs) or (CHWs) behind it
Response: the abbreviations were spelt out fully. Page 7, Line (266)
On Page 8 you wrote: For example, Community Health Valunteers CHVs could engage specific audiences like Boda Boda (motocycle) riders, adolescents, women, chamas, and community influencers in group discussions.
It was not until this page that I understood what a CHV is. Provide this definition and others on the previous page (Page 7):
Response: We have provided the definitions. Page 7, Line (266)
Also, change to: For example, Community Health Volunteers CHVs could engage specific audiences like Boda Boda (motocycle) riders, adolescents, women, chamas, and community influencers in group discussions.
Response: Thank you. We have edited as advised. Page 8, Line (298-301)
DISCUSSION
On Page 11 you wrote: Our respondents have similarly indicated the importance of community engagement, the use of influencers and role models, and encouraging stakeholders to work together, which is also supported by the literature as a study in thirteen African countries have emphasized the importance for African countries to address the challenges facing RCCE risk communication and community engagement to contain the pandemic effectively and to prepare for future public health emergencies [23]
Change to: Our respondents have similarly indicated the importance of community engagement, the use of influencers and role models, and encouraging stakeholders to work together, which is also supported by the literature as a study in thirteen African countries have emphasized the importance for African countries to address the challenges facing RCCE (risk communication and community engagement) to contain the pandemic effectively and to prepare for future public health emergencies [23]
Response: Thank you. We have edited as advised. Page 11, Line (256-260)
On Page 12, you wrote: The financial incentive method was highly criticized in the literature as several studies yielded that small financial incentives do not meaningfully improve COVID-19 vaccination rates among vaccine-hesitant [29,30].
Change to: The financial incentive method was highly criticized in the literature as several studies yielded that small financial incentives do not meaningfully improve COVID-19 vaccination rates among the vaccine-hesitant [29,30].
Response: Thank you. We have edited as advised. Page 11, Line (478-480)
On Page 12 you wrote: those marginalized populations can't leave their job to get vaccinated, and even if they did, they reach the closest vaccine center, when it is already closed.
Change to: those marginalized populations cannot leave their job to get vaccinated, and even if they did, they reach the closest vaccine center, when it is already closed.
- Do not use contractions in academic writing (unless it is a direct quote)
Response: Thank you. We have edited as advised. Page 12, Line (495-597)
On Page 12 you wrote: Results from another study on the influenza vaccine have indicated the essential role of increasing the number of training for the vaccine staff as this had shown to level up the influenza vaccine uptake [36].
Change to: Results from another study on the influenza vaccine have indicated the essential role of increasing the number of trainings for vaccine staff as this was shown to level up the influenza vaccine uptake [36].
Response: Thank you. We have edited as advised. Page 12, Line (497-499)
On Page 12 you wrote: The last theme in our results, which was supported by interviewees from rural areas but refused by interviewees from urban areas, is making vaccination mandatory.
Change to: The last theme in our results, which was supported by interviewees from rural areas but was not supported by interviewees from urban areas, is making the vaccination mandatory.
Response: Thank you. We have edited as advised. Page 12, Line (500-501)
On Page 12 you wrote: However, a different study concluded that herd immunity could be 558 reached without a policy of compulsory vaccination, none the less the 559 same study stated that mandatory policy might be found admissible, too, 560 were it to become critical [39].
- Define what you mean by “herd immunity”
Response: We have explained what we mean by ‘herd immunity’ Page 12, Line (507-508)
On Page 13, you wrote: this should be applicable only if benefits outweigh harm for HCWs, patients' welfare is enhanced, and fair rules and exemptions are defined [40].
Change to: this should be applicable only if the benefits outweigh harm for HCWs, patients' welfare is enhanced, and fair rules and exemptions are defined [40].
Response: Thank you. We have edited as advised. Page 12, Line (512-514)
On Page 13 you wrote: Second, although our researcher has conducted the necessary advanced trainings and experiences, the potential for researcher subjectivity and bias may have influenced the data collection, analysis, and interpretation processes.
- How did the researcher decrease bias?
Response: we added “Hence, we implemented measures like peer reviews, detailed notetaking, and transparent reporting to reduce likelihood of biases.”
Page 29, Line (539-540)
Other Issues:
- What is the theoretical framework on which your study is built? A theoretical foundation would make your work stronger.
Response: Thank you for this proposal. Nevertheless, the authors chose to focus attention on describing the study findings contained in Table 1.
On Page 11, you wrote: Doctors should be polite. Someone may be discouraged to visit a vaccination centre even if it's nearer if they find out that the health care workers there are rude. (Rabai, FGD 20-49) It is difficult because they [HCPs] are the ones who are supposed to guide us... However, I think the healthcare workers should be human. Some of them were even selling the vaccines. (Rabai, Teacher, IDI 8)
- Doctors need to treat patients with respect. They should not be rude (impolite). Discuss mandatory trainings that doctors should receive related to patient care.
Response: We added a paragraph in the discussion. Page 12, Line (515-522)
- Selling of vaccines is illegal and unethical. Discuss the specific actions that should be taken against healthcare workers that engage in this illegal and unethical practice.
Response: a small paragraph was added in the discussion. Page 12 Line (522-525)
While I found your table to be very succinct, I believe that adding a Conceptual Framework for your work would bring added depth.
Response: Thank you for this proposal. Nevertheless, the authors chose to focus attention on describing the study findings contained in Table 1.
Thank you,
AHMED ASA’AD AL-AGHBARI (On behalf of all authors)
